# Bi-parametric prostate MR image synthesis using pathology and sequence-conditioned stable diffusion

**Shaheer U. Saeed** [1]                                   SHAHEER.SAEED.17@UCL.AC.UK
**Tom Syer** [2]                                              T.SYER@UCL.AC.UK
**Wen Yan** [1,3]                                             WEN-YAN@UCL.AC.UK
**Qianye Yang** [1]                                         QIANYE.YANG.19@UCL.AC.UK
**Mark Emberton** [4]                                     M.EMBERTON@UCL.AC.UK
**Shonit Punwani** [2]                                     S.PUNWANI@UCL.AC.UK
**Matthew J. Clarkson** [1]                             M.CLARKSON@UCL.AC.UK
**Dean C. Barratt** [1]                                     D.BARRATT@UCL.AC.UK
**Yipeng Hu** [1]                                           YIPENG.HU@UCL.AC.UK

[1] *Centre for Medical Image Computing, Wellcome/EPSRC Centre for Interventional and Surgical Sciences and Department of Medical Physics and Biomedical Engineering, University College London, London, UK.*

[2] *Centre for Medical Imaging, Division of Medicine, University College London, London, UK.*

[3] *City University of Hong Kong, Department of Electrical Engineering, Hong Kong, China*

[4] *Department of Urology, University College Hospital NHS foundation Trust; and Division of Surgery and Interventional Science, University College London, London, UK.*

**Editors:** Accepted for publication at MIDL 2023

## Abstract

We propose an image synthesis mechanism for multi-sequence prostate MR images conditioned on text, to control lesion presence and sequence, as well as to generate paired bi-parametric images conditioned on images e.g. for generating diffusion-weighted MR from T2-weighted MR for paired data, which are two challenging tasks in pathological image synthesis. Our proposed mechanism utilises and builds upon the recent stable diffusion model by proposing image-based conditioning for paired data generation. We validate our method using 2D image slices from real suspected prostate cancer patients. The realism of the synthesised images is validated by means of a blind expert evaluation for identifying real versus fake images, where a radiologist with 4 years experience reading urological MR only achieves 59.4% accuracy across all tested sequences (where chance is 50%). For the first time, we evaluate the realism of the generated pathology by blind expert identification of the presence of suspected lesions, where we find that the clinician performs similarly for both real and synthesised images, with a 2.9 percentage point difference in lesion identification accuracy between real and synthesised images, demonstrating the potentials in radiological training purposes. Furthermore, we also show that a machine learning model, trained for lesion identification, shows better performance (76.2% vs 70.4%, statistically significant improvement) when trained with real data augmented by synthesised data as opposed to training with only real images, demonstrating usefulness for model training.

**Keywords:** Stable Diffusion, Image Synthesis, MRI, Prostate.

## 1. Introduction

Image synthesis has been used to augment real training data, for domain adaptation, or to generate training data for applications with limited real labelled data (Kazeminia et al., 2020). Within these applications, it has also been demonstrated to be effective for improving task performance of automated task networks for tasks like organ segmentation, when synthetic data is used in conjunction with real (Chartsias et al., 2017; Frid-Adar et al., 2018). While these applications do benefit from realistic synthesised data, it is important for these synthesised images to be conditioned on features that need simulating within the context of the relevant clinical application (Kazeminia et al., 2020; Skandarani et al., 2021).

Before the advent of deep learning and even to this day, physics- or pre-operative-imaging-based simulators have been proposed for medical image synthesis, inspired by the physics of the underlying imaging processes. An example is generating intra-operative ultrasound data from larger and higher-resolution pre-operative images such as CT (Cong et al., 2013; Shams et al., 2008). Machine learning approaches for medical image synthesis have largely been focused on using generative adversarial networks (GANs) (Kazeminia et al., 2020; Skandarani et al., 2021; Singh and Raza, 2021). Conditional GANs (cGAN) have been applied to a variety of problems, for both pre- and intra-operative image synthesis, without relying on patient anatomy being available for synthesis (Kazeminia et al., 2020; Skandarani et al., 2021; Singh and Raza, 2021). They have also been used for learning inter-modality correspondence e.g. for generating CT from MR or vice versa (Nie et al., 2017; Wolterink et al., 2017a,b; Chartsias et al., 2017), conditioned image synthesis for features of interest e.g. blood vessels (Costa et al., 2017a,b; Isola et al., 2017; Guibas et al., 2017; Zhao et al., 2017) or tumours (Mukherkjee et al., 2022; Li et al., 2020). While these GAN-based methods produce realistic images, they often suffer from problems such as under-represented features of interest (Kazeminia et al., 2020), poor performance on class-imbalanced data-sets (Kazeminia et al., 2020), especially for under-represented classes, or other common problems encountered during training, e.g. unstable training, mode collapse and diminishing gradients (Saxena and Cao, 2021), which prevent them from being widely usable (Li et al., 2021). Our preliminary experiments using cGANs or their variants were consistent with these identified limitations, with unconvincing results such as prostate gland broken, in-painted or lacking details, see illustrated examples (Fig. 1). The results may suggest lack of generative modelling ability or ineffective conditioning, for this challenging application with often subtle and sometimes radiologically undetermined pathology.

Recently, diffusion probabilistic models (DPM) have been proposed for image synthesis (Ho et al., 2020; Nichol and Dhariwal, 2021). These models have shown improved image synthesis capability for a wide array of problems and are capable of incorporating both image-based and text-based conditioning (Ho et al., 2020; Nichol and Dhariwal, 2021; Ramesh et al., 2022; Saharia et al., 2022; Rombach et al., 2022). While these models have been leveraged for medical image synthesis tasks (Kazerouni et al., 2022; Kim and Ye, 2022; Dorjsembe et al., 2022; Moghadam et al., 2023; Pinaya et al., 2022), studies so far have focused mostly on unconditional image synthesis or synthesis conditioned on text-based variables.

In many applications that estimate clinically important pathology, conditioning on often under-represented diverse pathology or less-specific, higher dimensional data, such as a specific image sequence in multi-parametric prostate MR, is beneficial or even inevitably

required. The ability to generate pathological images potentially can provide useful data, for training junior clinicians as well as developing machine learning models, partly caused by the low sample availability to diversity ratio in specific disease conditions. In addition, in our application, prostate cancer is potentially sensitive to multiple sequences of MR images, reflected by modern uro-radiological guidelines such as PiRADS (pir). For example, peripheral zone lesions are primarily graded on DW images, while transition zone cancers are predominantly determined on T2 images but their risk may be upgraded with positive findings on DW images. The ability to model the conditional distribution of these paired image data and to synthesise diffusion images, given other complementary sequences are essential for the above-discussed data augmentation and clinician training applications.

In this work we present a conditional DPM for synthesis of prostate MR images, conditioned not only on text to control variables such as the presence of cancer and sequence of MR acquisition, i.e. T2-weighted (T2W), apparent diffusion contrast (ADC), diffusion-weighted (DW), but on images to facilitate generation of corresponding image pairs by generating DW images conditioned on T2W images. Compared to previous work (Rombach et al., 2022; Saharia et al., 2022; Ramesh et al., 2022), our work investigates combined conditional image- and text-based synthesis as opposed to only text-based or unconditional synthesis, or image modification. Furthermore, the text-conditioning used in our work represents pathological status as opposed to other common visual descriptors used in previous works. Image-based conditioning for synthesis of MR sequences is also novel, and different from the previously proposed diffusion-based image modification e.g. inpainting or super-resolution. We conduct evaluation by: 1) presenting images to a clinician to identify synthesised images, to test the efficacy of the image synthesis, to demonstrate the realism of the synthesised images; 2) presenting corresponding T2-weighted and diffusion-weighted, generated and real, images to a clinician for a cancer detection task, to test the realism of the generated lesions; and 3) testing the segmentation accuracy for a neural network-based multi-parametric MR lesion identification task with real data versus with a dataset augmented using synthesised images.

The contributions of this work are summarised: 1) We propose a DPM for synthesis of prostate MR images with challenging pathology. 2) We propose conditioning the synthesis on text-based inputs to control presence of lesions and MR sequence. 3) We propose conditional image synthesis to generate DW images from corresponding T2W images as a means of generating corresponding multi-sequence images. 4) We conduct an evaluation to demonstrate the effectiveness of the synthesised images not only for clinical training use cases but also for machine learning model training for a task carried out with the MR images i.e. lesion identification.

## 2. Methods

### 2.1. Forward process

In our application, assume a forward process being the addition of noise in sequential steps to a given input image, $\mathbf{x}_0 \sim q(x|y)$ sampled from a data distribution of real samples $q(x)$, the distribution to be modelled. Here, $y$ is a variable used to condition the data. At each time-step $t$, where $t \in \{1, T\}$, the added Gaussian noise follows a Markov chain with $T$ steps, with variance $\beta_t$, and only depends on the sample from the previous step and the

variable used for conditioning. The distribution can then be written as $q(\mathbf{x}_t|\mathbf{x}_{t-1}, y)$, with $\mathbf{x}_t$ as a latent variable. The diffusion forward process can, therefore, be formulated as:

$$q(\mathbf{x}_t|\mathbf{x}_{t-1}, y) = \mathcal{N}(\mathbf{x}_t; \mu_t = \sqrt{1-\beta_t}\mathbf{x}_{t-1}, \sigma_t = \beta_t \mathbf{I}) \tag{1}$$

Going from the sample $\mathbf{x}_0$ to the sample $\mathbf{x}_T$ in a closed form thus looks like:

$$q(\mathbf{x}_{1:T}|\mathbf{x}_0, y) = \prod_{t=1}^{T} q(\mathbf{x}_t|\mathbf{x}_{t-1}, y) \tag{2}$$

Efficient re-parameterising allows to compute $\mathbf{x}_t$ without needing to compute all the samples at previous steps. As detailed in previous works (Ho et al., 2020; Nichol and Dhariwal, 2021; Rombach et al., 2022), it can be shown that by defining $\alpha_t = 1 - \beta_t$, $\bar{\alpha}_t = \prod_{s=0}^{t} \alpha_s$, a sample $\mathbf{x}_t$ may be sampled as follows:

$$\mathbf{x}_t \sim q(\mathbf{x}_t|\mathbf{x}_{t-1}, y) = \mathcal{N}(\mathbf{x}_t; \sqrt{\bar{\alpha}_t}\mathbf{x}_0, (1-\bar{\alpha}_t)\mathbf{I} \tag{3}$$

Variance $\beta$ may be fixed, whilst a cosine schedule (Nichol and Dhariwal, 2021) is adapted in this work, increasing $\beta$ from $10^{-4}$ to $0.02$ over $T$ steps.

## 2.2. Additional input $y$ for conditioning

As in Rombach et al. (2022), we use an encoder $\tau_\gamma$, with weights $\gamma$, for our variables used to condition the synthesis $y$, which maps to an intermediate representation $\tau_\gamma(y) \in \mathbb{R}^{M \times d_\tau}$, which is then mapped to the intermediate layers of the U-Net. For language prompts, the domain-specific encoder may be a transformer model as in Rombach et al. (2022). For higher-dimensional conditioning data, such as an image, we use a latent representation from a trained auto-encoder to generate the encoding. In this work the auto-encoder is implemented as a convolutional neural network with 3 down-sampling and 3 up-sampling layers with the latent representation being 128-dimensional; this is mapped to the intermediate layers of the U-Net, similar to Rombach et al. (2022). For sample $\mathbf{x}_t$, conditioned on $y$, consisting of the image encoding and text encoding, this gives us $\mathbf{x}_t \sim q(\mathbf{x}_t|\mathbf{x}_{t-1}, \tau_\gamma(y))$. For notational brevity, however, we use $y$, for both, in the remaining analysis.

## 2.3. Reverse process

With a sufficiently large $T$, the distribution approaches an isotropic Gaussian (Ho et al., 2020). Thus, the data distribution $q(x)$ can be modelled by reversing the noise adding process from unit Gaussian distribution $\mathcal{N}(0, \mathbf{I})$ samples. In practice, however, the reverse $q(\mathbf{x}_{t-1}|\mathbf{x}_t, y)$ is not known or can be statistically estimated since any statistical estimates would involve knowing the data distribution (Ho et al., 2020). We can, however, learn to approximate $q(\mathbf{x}_{t-1}|\mathbf{x}_t, y)$ using a parameterised function $\epsilon_\theta(\mathbf{x}_t, t, y)$ which can be interpreted as a sequence of de-noising auto-encoders with additional conditioning on the time-step $t$. It is easier to parameterise a Gaussian and then remove the predicted Gaussian noise manually. Thus for sample $\mathbf{x}_{t-1}$ we have:

$$p_\theta(\mathbf{x}_{t-1}|\mathbf{x}_t, t, y) = \mathcal{N}(\mathbf{x}_{t-1}; \mu_\theta(\mathbf{x}_t, t), \sigma_\theta(\mathbf{x}_t, t)) \tag{4}$$

Then applying the reverse process for all time-steps:

$$p_\theta(\mathbf{x}_{0:T}, 0:T, y) = p_\theta(\mathbf{x}_T, T, y) \prod_{t-1}^{T} p_\theta(\mathbf{x}_{t-1}|\mathbf{x}_t, t, y) \tag{5}$$

Here, we summarise the entire process as a de-noising function $\epsilon_\theta(\mathbf{x}_t, t, y)$ which is trained to predict a de-noised version of its input i.e. $\mathbf{x_0}$ from $\mathbf{x}_t$. To incorporate the encoder used to pre-process the conditioning variable we may re-write this as $\epsilon_\theta(\mathbf{x}_t, t, \tau_\gamma(y))$. As shown in previous works (Ho et al., 2020; Nichol and Dhariwal, 2021; Rombach et al., 2022), the objective for this can then be simplified to:

$$\mathcal{L} = \mathbb{E}_{x,\epsilon \sim \mathcal{N}(0,1),t,y} \left[ ||\epsilon - \epsilon_\theta(\mathbf{x}_t, t, y)||_1^1 \right] \tag{6}$$

In this work, we follow the implementation in Rombach et al. (2022) and use a U-Net as our de-noising function and feed latent representations to the function for training, as in Rombach et al. (2022), for computational efficiency.

### 2.4. Sampling from the trained diffusion model

To sample an image from the learnt data distribution with the given conditioning variables. We sample $\mathbf{x}_T \sim \mathcal{N}(0, \mathbf{I})$ and compute the sample using our reverse de-noising function $\mathbf{x}_0 = \epsilon_\theta(\mathbf{x}_t, t, \tau_\gamma(y))$. In practice, however, better performance is seen when noise is added back using the noise schedule until step $t - 1$, i.e. using Eq 2.1, and then the de-noising function is applied again to generate another latent sample $\mathbf{x}_0$ to which noise is added back using the schedule until $t - 2$, repeated until $t - t$ (Rombach et al., 2022).

## 3. Experiments

### 3.1. Datasets

We used two datasets for training, a large open dataset to train and initialise the model for training with a smaller dataset, both of which are described below.

**Open-source prostate MR data** This dataset comes from Saha et al. (2022) with 1285 samples of 3D prostate MR images with T2W and ADC available. 190 samples were removed after a semi-manual process, due to questionable quality and unclear conformity to radiological reporting standards. Only 10 central slices were used for training from each of the 3D images since they contain the majority of the prostate volume. 200 patient cases were held out from training and used for evaluation, details in Sec. 3.3. Further details in Saha et al. (2022). This dataset was used to generate ADC and T2W used in Sec. 3.3.

**Closed source multi-sequence prostate MR data** Multi-parametric 3D MR images were acquired from 850 patients undergoing prostate biopsy and therapy as part of trials at University College London Hospitals (Hamid et al., 2019; Simmons et al., 2018; Bosaily et al., 2015; Orczyk et al., 2021; Dickinson et al., 2013; Linch et al., 2017). Patients gave written consent and ethics approval was obtained as part of the respective trial protocols. Lesions were manually contoured by a radiologist and were used to generate slice-level binary labels of lesion presence. Image sequences included T2W and DW with highest b-values =

1000 or 2000. For the purpose of this study, ten central slices were used for training from each of the 3D images since they contain the majority of the prostate volume. Slices with lesions visible, PIRADS $\geq 3$, are marked as containing a lesion. 200 samples were held out from training for purposes of evaluation, see details in Sec. 3.3. This dataset was used to generate paired T2W and DW images as described in Sec. 3.3. It should be noted that after training the model with the open-source dataset described above, the model was only fine-tuned with the paired closed source data.

## 3.2. Model implementation and training

We closely follow the implementation used in Rombach et al. (2022). Hyper-parameters are empirically configured, remaining the same as ImageNet experiments using latent diffusion model or LDM-1 from Rombach et al. (2022) (Appendix). Networks are trained by first setting to pre-trained weights and the BERT tokeniser is used together with the provided encoder for encoding text prompts Rombach et al. (2022) (Appendix). The encoder for cross-sequence translation, i.e. T2W to DW, is trained as a convolutional auto-encoder with latent representations used for conditioning, details in Sec. 2.2. Models in this work were trained on slice-level 2D data, for computational/ development efficiency, utilising robust training strategies and hyperparameter values which may not generalise to 3D samples.

## 3.3. Usability study

**Expert identification of synthesised ADC and T2W images**  In this experiment independent 2D slices of T2W and ADC, which are not necessarily from the same patient, are used. We ask the clinician with 4 years experience reading urological MR, to identify synthesised images from a mixture of real and synthesised images, regardless of images containing lesions. Data used consists of 32 2D ADC and 32 2D T2W image slices containing the prostate, with equal positive-to-negative ratio for lesions and real-to-synthesis ratio. The ratios are also blind to the observer. Comparisons and results reported in Sec. 4.

**Expert identification of lesions on ADC and T2W images**  Using the same data as in the above synthesised identification task, we now ask the clinician to identify 2D slices that contain a suspected lesion with PIRADS $\geq 3$, regardless of the images being real or fake. Comparisons and results reported in Sec. 4.

**Expert identification of lesions on paired T2W-DW images**  Using paired T2W-DW data, the clinician was asked to identify images with a suspected lesion, PIRADS $\geq 3$, regardless of images being real or synthesised. Synthesised paired images were generated by conditioning DW generation on T2W images. The 32 2D paired T2W- DW images used for this study are also sampled with equal ratios, between positive and negative and between real and synthesised, both blind prior to the observer, results reported in Sec. 4.

**Machine learning-automated lesion detection**  An AlexNet (4 convolutional and 4 fully connected layers) was trained for the binary classification task of lesion identification, as performed by an expert in the previous paragraph, on each 2D slice pair of T2W and DW images, at the same depth from the same subject. We note that the performance reported in Sec. 4 is consistent with those reported in similar applications, e.g. (Kwon et al., 2018),

| Task | Data | Accuracy |
|---|---|---|
| Clinician - synthesised identification
$N_{\text{train}} = 8950$
$N_{\text{usability study}} = 32$ T2W, $32$ ADC | ADC (overall) | 0.563 |
| | ADC (w/ cancer) | 0.625 |
| | ADC (w/o cancer) | 0.500 |
| | T2W (overall) | 0.625 |
| | T2W (w/ cancer) | 0.688 |
| | T2W (w/o cancer) | 0.563 |
| Clinician - lesion identification
$N_{\text{train}} = 8950$
$N_{\text{usability study}} = 32$ T2W, $32$ ADC | ADC (overall) | 0.688 |
| | ADC (real) | 0.625 |
| | ADC (synthesised) | 0.750 |
| | T2W (overall) | 0.594 |
| | T2W (real) | 0.625 |
| | T2W (synthesised) | 0.563 |
| Clinician - lesion identification
$N_{\text{train}} = 6500$
$N_{\text{usability study}} = 32$ T2W-DW | T2W-DW paired (overall) | 0.563 |
| | T2W-DW paired (real) | 0.563 |
| | T2W-DW paired (synthesised) | 0.563 |
| ML model - lesion identification | Trained with real | $0.704 \pm 0.035$ |
| | Trained with real + synthesised | $0.762 \pm 0.042$ |

Table 1: Results for both expert clinician and machine learning model experiments, with details described in Sec. 3. '$N_{\text{set}}$' indicates the set size in terms of images.

therefore justifies the choice of the widely established baseline, which is also competitive in a wide variety of other tasks (Krizhevsky et al., 2012). Separate models were trained using only real images and a dataset augmented with the DPM synthesised images. For the real case, we use the closed-source dataset and split it into train, validation and holdout sets, with 510, 170 and 170 patients, respectively, resulting in a total of 5100, 1700 and 1700 2D slices in respective sets. For the augmented dataset, 1600 synthesised images with lesions and 1600 without, were split into train and validation sets. These were added to the original sets, resulting in 7500, 2500 in each of the sets, respectively. The holdout set remains the same with 1700 images. The classification accuracy results are reported on 1700 real 2D slices from the holdout set for both trained models, to compare the difference between training with and without augmented synthesised images, in Sec. 4.

## 4. Results

Results comparing real images to synthesised ones are presented in Table 1 and Fig. 1. The stable diffusion model took approximately 8 days to train on a single Nvidia Tesla V100 GPU, with an average time for sample synthesis being 10 seconds on the same GPU.

**Comparing real and synthesised images by expert observer** As shown in Table 1, an expert clinician was only able to identify synthesised images from real ones with an average accuracy of 0.594 averaged over all ADC and T2W images (where chance is 0.500).

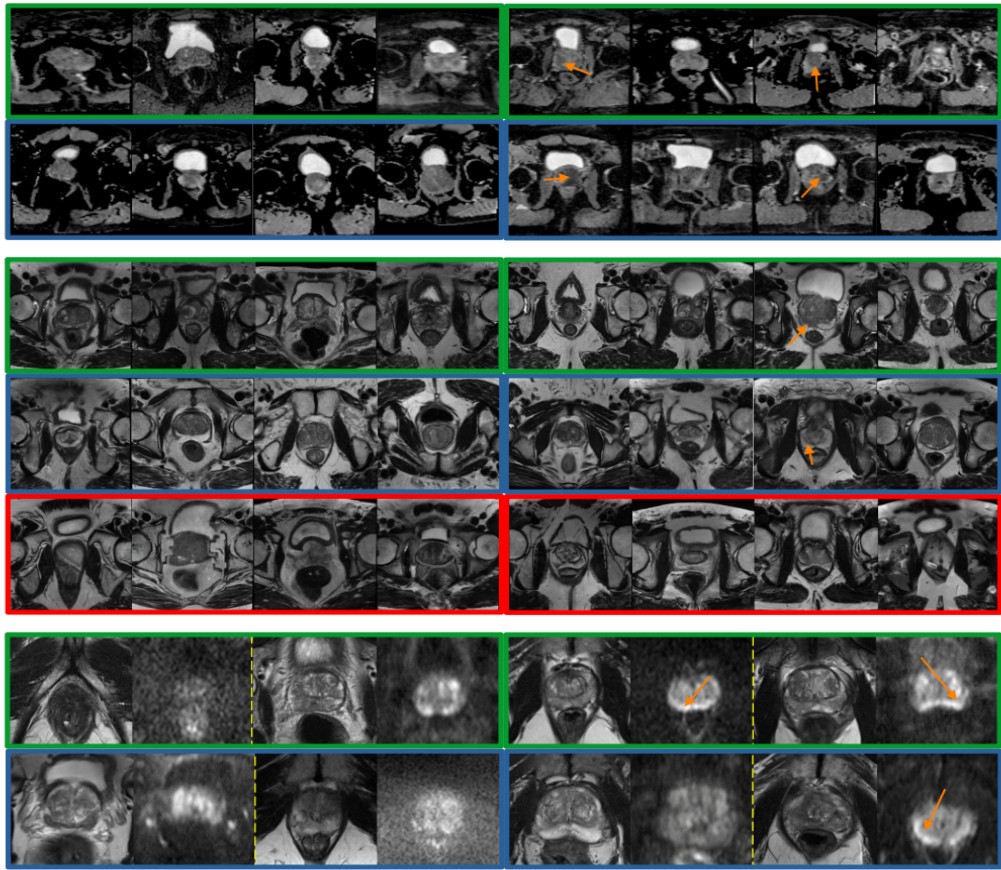

Figure 1: Examples of generated and real images, with keys as follows. Blue: Synthesised using DPM, Green: Real, Red: Synthesised using cGAN. Left: No Cancer, Right: Cancer (arrows indicating suspected lesions, w.r.t. PIRADS≥3). Top block: ADC, Middle block: T2W, Bottom block: Paired T2W-DW (left-right).

**Evaluating lesion realism by comparing expert lesion identification performance**
We compare the lesion identification task performance accuracy for real versus synthesised images. The accuracy for real images, averaged over ADC, T2W and paired T2W-DWI lesion identification as performed by the expert is 60.4%, and for synthesised averaged over the same is 62.6%. This gives us only a small difference of 2.1 percentage points between the lesion identification performance of the expert for real versus synthesised images.

**Evaluating usability of synthesised data for machine learning model training**
As in Table 1, we observe a 5.8% improvement in accuracy, with statistical significance (p-value=0.004), for the lesion identification task for the model trained with augmented data, compared to a model trained with only real images, as described in Sec. 3.3.

## 5. Discussion and Conclusion

Based on the results presented in Sec. 4, given that 1) the expert clinician only performed marginally better than chance in identifying synthesised images and 2) the expert lesion identification accuracy was similar between the real and generated data, we argue that this may enable a number of applications including flexible training tools for clinical trainees. Perhaps more interestingly, some of the most challenging radiologist tasks, such as the DW reading following a positive reading of T2W images for a transition zone lesion, could be simulated using the proposed sequence-conditioned models. Furthermore, performance improvement in lesion identification experiments using machine learning models, with and without adding synthesised data for training, concludes that the data synthesis approach is promising for data augmentation. Our proposed method is currently limited in terms of which areas of prostate can be synthesised since mostly central parts are synthesised by the trained network, partly due to random sampling without explicit positional conditioning; we plan to investigate further conditioning schemes that allow control over regions of the prostate that need to be synthesised. E.g., conditioning mechanism to specify the location (apical vs near base, peripheral vs transition zones) and severity (PIRADS scores) of lesions. This would indeed be interesting for targeted sample generation for both applications.

This work proposes a stable-diffusion-based image synthesis method together with a conditioning scheme to generate realistic prostate MR images where the sequence of MR, presence of lesions and synthesis of paired data may be controlled. Our quantitative results demonstrate the usability of the generated images for use cases such as training trainee clinicians/ radiologists and for improving machine learning model task performance.

## Acknowledgements

This work was supported by the EPSRC [EP/T029404/1], Wellcome/EPSRC Centre for Interventional and Surgical Sciences [203145Z/16/Z], and the International Alliance for Cancer Early Detection, an alliance between Cancer Research UK [C28070/A30912; C73666/A31378], Canary Center at Stanford University, the University of Cambridge, OHSU Knight Cancer Institute, University College London and the University of Manchester.

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

## Appendix

### Hyper-parameters

The hyper-parameters used in our study are specified in Tab. 2. Training was stopped after convergence, which is defined as observing an improvement less than $1e^{-6}$ in the loss, which is sometimes referred to as the minimum-delta.

| Hyperparameter | Value |
|---|---|
| Diffusion steps (train) | 1000 |
| Number of parameters | 396M |
| Channels | 192 |
| Depth | 2 |
| Channel Multiplier | 1, 1, 2, 2 , 4, 4 |
| Batch Size | 7 |
| Embedding Dimension | 512 |

Table 2: Hyperparameters used for training the diffusion model which are set empirically based on experiments from Rombach et al. (2022).

### Examples of text prompts

Examples of text prompts used for training are presented in Tab. 3. We used the presence of cancer together with MR sequence to form text prompts. The prompts were generated randomly from a list of 8 phrases with the MR sequence or lesion presence being inserted into each of the phrases as appropriate. Due to the vast prior research into BERT-based text encoding for diffusion models, we chose to opt for BERT to generate our text encodings. This allows us to avoid training a binary variable encoder which can map to the intermediate layers of the UNet, which may be difficult since auto-encoders have not demonstrated to be suitable to map to dimensions higher than the input as they mostly rely on bottlenecks with lower dimensions compared to the input for encoding.

| Text prompts |
|---|
| 'A T2 image of a prostate with a lesion' |
| 'Prostate DW image with a lesion' |
| 'ADC image of a prostate with no lesion' |
| 'Image of ADC prostate without lesion' |

Table 3: Examples of text prompts used

Variability in these text prompts may promote learning higher-level concepts such as 'prostate', 'lesion' etc.

### Methods overview

An overview of the diffusion model is presented in Fig. 2. For obtaining a sample, we iterate this 50 times by first generating $x_T$ randomly, obtaining an estimate for $x_0$ using

the reverse diffusion network and then adding back 49/50-th of the noise back. Then we repeat reverse diffusion and add back 48/50th of the noise. We run this for 50 steps until we add back 0/50-th of the noise and obtain our final sample.

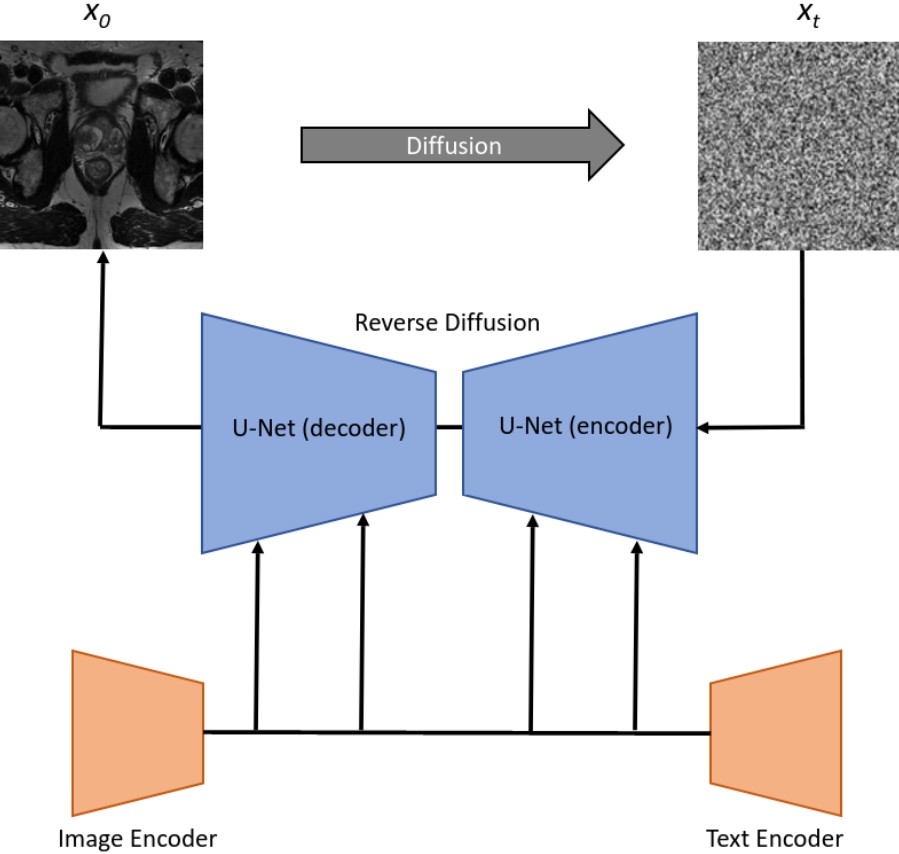

Figure 2: An overview of the diffusion process.

