# OpenReview forum: "Bi-parametric prostate MR image synthesis using pathology and sequence-conditioned stable diffusion"
_MIDL.io/2023/Conference — MIDL 2023 Oral_

### Official Review · Reviewer_4gLi · 2023-01-29

**Confidence:** 4
**Preliminary Rating:** 4

**Summary:**

This paper applied the recent stable diffusion model for multi-modality prostate MR images conditioned on text, to control lesion presence and modality, as well as to generate diffusion-weighted MR from T2-weighted MR for paired data. The validation utilizes a blind expert evaluation for identifying real versus fake images.

**Strengths:**

A timely study on applying the recent conditional diffusion probabilistic models in typical computer vision to the prostate MRI synthesis task. From this perspective, the application is interesting.
Introducing the expert evaluation is good and makes sense.


**Weaknesses:**

Authors should further polish the paper, especially by giving some illustrations to help readers get the spirit and avoid many wordy expressions. For example, an illustration for the conditioned DPM, considering that many details have been omitted here due to page limit.

Regarding the technical perspective, please highlight the contribution and difference between your work and (Rombach et al. 2022).
Rombach, Robin, et al. "High-resolution image synthesis with latent diffusion models." Proceedings of the IEEE/CVF Conference on Computer Vision and Pattern Recognition. 2022.

How many clinicians are involved in your evaluation? Inter-clinician variation is important to discern the effectiveness, e.g., what is the std?

The quantitative comparison with other methods seems missing like cGAN or other text-conditional image synthesis methods mentioned in (Rombach et al. 2022). Otherwise, it is hard to discern the superiority of this method.


**Deanonymize Review:**

no

**Paper Type:**

methodological development

**Questions To Address In The Rebuttal:**

See weakness.

Authors should further polish the paper, especially by giving some illustrations to help readers get the spirit and avoid many wordy expressions. For example, an illustration for the conditioned DPM, considering that many details have been omitted here due to page limit.

Regarding the technical perspective, please highlight the contribution and difference between your work and (Rombach et al. 2022).
Rombach, Robin, et al. "High-resolution image synthesis with latent diffusion models." Proceedings of the IEEE/CVF Conference on Computer Vision and Pattern Recognition. 2022.

How many clinicians are involved in your evaluation? Inter-clinician variation is important to discern the effectiveness, e.g., what is the std?

The quantitative comparison with other methods seems missing like cGAN or other text-conditional image synthesis methods mentioned in (Rombach et al. 2022). Otherwise, it is hard to discern the superiority of this method.

---

### Official Review · Reviewer_vsJu · 2023-02-04

**Confidence:** 4
**Preliminary Rating:** 5
**Recommendation:** Oral, Poster

**Summary:**

This paper is making use of the latest developments in latent diffusion models in order to perform fairly challenging image generation tasks in the area of prostate MRI. The trained models are able to generate axial prostate MR slices with different contrasts (ADC, T2 and diffusion weighted), with and without (prostate) lesions. The contrast and lesion presence are controlled via a text sequence that is input into the model (and encoded via BERT). Additionally, a convolutional AE's latent representation is used to condition diffusion weighted images on a T2 weighted input in order to produce paired data.

**Strengths:**

The paper is fairly well-written and applies very recent developments to an image generation problem that is argued to be of relevance e.g. for training purposes (both for doctors as well as for algorithms), as is already pointed out in the abstract.  The two datasets employed are reasonably large (at least in terms of patients).

**Weaknesses:**

The presented approach uses a BERT language model in order to condition the generated images, but it is not explained at all what kind of information is input as text sequence. No example is given, and it is not stated whether the information is just the type of MR contrast and the binary presence of tumors (in which case the reader may wonder if a language model is necessary in the first place), or whether the information is a radiological report with lesion location, size, or other characteristics.

The manuscript explains the basics of DPMs, but it does not give many details on the architecture or the training procedure of the proposed model. It mainly refers to the refer to the CVPR 2022 paper by Rombach et al., but even if the code and architecture was used unchanged, the fine-tuning procedure should be covered (stop criterion, for instance). Putting this information into an appendix could be an idea, if space limits are the reason, but it would help reproducibility.

The results are presented in a biased way, unnecessarily making the error measures appear lower than they are.  See detailed comments.

**Deanonymize Review:**

yes

**Detailed Comments:**

The manuscript uses the unit "%" in a misleading way when it would be more appropriate to use percent *points* ("p.p.", if you want). "59.4% accuracy" is not "9.4% better than chance", although that may sound good. I am not sure that "18.8% better than chance" (which one could also claim to be mathematically correct) would be better, though, because I would not measure "better" in % of the expected 50% accuracy.

The statement on the lesion realism "By comparing the lesion identification task performance accuracy for real and synthesised images, we observe only a small difference of 2.1% between the accuracy for real images versus for synthesised images, ..." is very hard to follow. First of all, the expert's accuracy of the discrimination between real and synthesised is 12.5 percent points higher for both ADC and T2W for images with cancer, which could be attributed to the lesion realism. Second, the accuracy for the lesion detection task (which may be even less clearly relevant to the lesion realism) is *very* different for the different contrasts, so that averaging does not appear to be well motivated.

This may be debatable, but I would prefer a different term than "modality" for different MR contrasts / protocols / sequences.  (I think there is some ambiguity on whether "modality" is on the level of MRI/CT/US/microscopy, or whether it can also apply to different contrast mechanisms.)

Figure 1 is very useful, but I suggest to redo it a) with properly aligned borders, and b) with different colors that are easier to relate to the semantics – for instance, green may be good for the real images / ground truth, whereas red may be best for the "bad" cGAN results.

One confusing sentence that I had to read multiple times can probably be shortened, simplified, and clarified at the same time: "For the augmented dataset, 1600 synthesised images with lesions and 1600 without, were split into train, validation sets, and added to the original sets, resulting in 7500, 2500 in each of the sets, respectively."

Other phrases that should be checked:
- "we ask the clinician, with 4 years experience" ("the", comma)
- "As in Table 1, ..."

**Paper Type:**

methodological development

**Questions To Address In The Rebuttal:**

I would like my question about the text input to be answered (see "Weaknesses").

The "Datasets" section states that only the central 10 slices of the MR volumes are used, since these contain the "majority of the prostate volume". I wonder whether this means that the cranial and caudal ends of the prostate gland are part of the training set and can be generated by the resulting model – although I am not a prostate expert, I think apical lesions are not uncommon or unimportant.

---

### Official Review · Reviewer_s4ud · 2023-02-05

**Confidence:** 3
**Preliminary Rating:** 5
**Recommendation:** Poster

**Summary:**

image synthesis mechanism for multi-modality prostate MR images con- ditioned on text, to control lesion presence and modality, as well as to generate paired bi-parametric images conditioned on images e.g. for generating diffusion-weighted MR from T2-weighted MR for paired data.

Proposed mechanism utilises and builds upon the recent stable diffusion model by proposing image-based conditioning for paired data generation.

**Strengths:**

Interesting application of Robin Rombach, et al paper on "High-resolution image synthesis with latent diffusion models" in IEEE/CVF CVPR conference, 2022.

High quality data set and extensive evaluation of multiple applications of clinical interest.

=================================================================

**Weaknesses:**

- "Slices with lesions visible, being PIRADS ≥ 3, are marked as containing a lesion.": how limiting is this toward clinical impact?
- Table 2 text: Left column, add training and test sizes (N=?) in the table.
- "5. Discussion and Conclusion": add some limitations to your study and views on future work.


=================================================================

**Deanonymize Review:**

no

**Paper Type:**

validation/application paper

**Questions To Address In The Rebuttal:**

Cf weaknesses:

- "Slices with lesions visible, being PIRADS ≥ 3, are marked as containing a lesion.": how limiting is this toward clinical impact?
- Table 2 text: Left column, add training and test sizes (N=?) in the table.
- "5. Discussion and Conclusion": add some limitations to your study and views on future work.

---

### Meta-Review · Area_Chair_cbZg · 2023-02-22

**Recommendation:** Accept (Oral)
**Confidence:** 4

**Metareview:**

Based on the reviewers' comments and my own reading, this is a strong paper with multiple strengths:
1. Uses the recent DPM models
2. Has extensive evaluations from multiple angles, including radiologists' evaluations, and ML based downstream tasks

I think it would be of significant interest to the community.